# A Unified Representation Learning Framework for Functional, Temporal, and Irregular Data

## Abstract

This paper presents a unified framework for learning representations from data that combine three challenging aspects: continuous functional structure, discrete temporal dynamics, and unstructured vector inputs. Our proposed models extend attention mechanisms to handle entire functions along a continuous axis, capture evolution over time, and integrate irregular or unordered features through modality fusion. This design enables the network to respect smoothness, temporal dependence, and heterogeneity while learning unified representations across modalities. As a motivating application, we study yield curve forecasting, where inputs may be full yield curves or bond-level trade data and outputs are predicted curves. Beyond forecasting, the framework is broadly applicable; for instance, to reinforcement learning tasks where policies evolve over time, or to generative models that produce structured functional outputs. Empirical results on bond datasets show consistent gains over classical econometric models, underscoring the potential of our approach as a general blueprint for representation learning with functional-temporal-unstructured data.

## 1 Introduction

Many modern machine learning problems involve data that are naturally expressed as *functions* rather than fixed-dimensional vectors. Examples arise in finance (yield curves, volatility surfaces, intraday trajectories), in engineering (stress-strain profiles, vibration signatures), in operations research (demand trajectories, cost surfaces), and in scientific computing (solutions to PDEs). Functional data analysis (FDA) provides a principled framework for modeling such data as continuous objects rather than discretized vectors; see Ramsay & Silverman (2005); Horváth & Kokoszka (2012). Classical approaches such as spline smoothing McCulloch (1975) and functional principal component analysis Rice & Silverman (1991); Hays et al. (2012) preserve smoothness but remain limited in handling nonlinearities or integrating exogenous information.

The yield curve offers a motivating example. It describes the relation between interest rates and time-to-maturity and plays a central role in fixed-income pricing, risk management, and macroeconomic policy. Econometric models, such as the Nelson–Siegel–Svensson (NSS) models Nelson & Siegel (1987); Svensson (1994) and their dynamic extensions Ang & Piazzesi (2003); Diebold & Li (2006), are widely used, but their parametric assumptions limit flexibility when faced with high-dimensional or irregular data.

Recent neural approaches generalize FDA to the deep learning setting. Functional neural networks extend multilayer perceptrons to function-valued inputs Thind et al. (2023); Wang & Cao (2025), while implicit neural representations encode continuous functions into network weights Dupont et al. (2022); Zhou et al. (2023); Tran et al. (2025), enabling end-to-end optimization without fixed basis expansions. These advances preserve functional structure, but most operate in isolation and cannot natively combine functional data with auxiliary, heterogeneous inputs such as bond-level trades.

In parallel, attention mechanisms have transformed representation learning. Transformers Vaswani et al. (2017) have become a foundation across language, vision, and multimodal learning Dosovitskiy et al. (2021). Extensions to scientific computing include operator-learning architectures such as DeepONets Lu et al. (2021), GNOT Hao et al. (2023), PiT Chen & Wu (2024), and functional-

attention hybrids Hong et al. (2024); Tran et al. (2025). These developments show how attention can act on continuous or function-valued objects, but they do not directly address multimodal inputs that combine functional trajectories, temporal dynamics, and unordered vector-valued sets.

This paper addresses the gap between FDA and modern transformer architectures by proposing a unified representation learning framework. The model extends attention mechanisms to function-valued inputs and supports joint integration of historical functional observations with irregular auxiliary vectors, such as bond-level market features. Our motivating application is yield curve forecasting with China Development Bank bond data, but the framework applies more broadly across finance, engineering, operations research, and scientific computing Chen & Wu (2024); Tran et al. (2025), offering a general blueprint for learning from functional, temporal, and irregular data.

**Contributions.** Our contributions are threefold. First, we introduce a transformer-style architecture that directly learns from functional time series, preserving their continuity and structural properties. Second, we generalize self- and cross-attention to operate on function-valued inputs, enabling principled representations of infinite-dimensional objects. Third, we design mechanisms to incorporate irregular, unordered auxiliary vectors, such as varying numbers of bond trades or heterogeneous sensor inputs, through masking and set-based embeddings. Empirically, we demonstrate the effectiveness of this framework on yield curve forecasting, where it outperforms classical econometric models, highlighting its promise as a general tool for representation learning beyond finance.

## 2 PROBLEM FORMULATION

At each time $t$, we observe two types of data: a function $Y_t : \mathcal{T} \mapsto \mathbb{R}$, where $\mathcal{T} \subset \mathbb{R}$ denotes the function domain, and a variable-sized set of vectors $X_t = \{x_t^{(1)}, \ldots, x_t^{(n_t)}\}$, which encodes additional information available at time $t$ with $x_t^{(i)} \in \mathbb{R}^m$. The goal is to learn a predictive model that integrates the temporal structure of both modalities to forecast the future functional response. Formally, given past $M$-step observations $\{(Y_s, X_s)\}_{s=t-M}^{t-1}$, the task is to predict the function at the next timestep:

$$\widehat{Y}_t = \hat{f}\Big(\{Y_s\}_{s=t-M_Y}^{t-1}, \{X_s\}_{s=t-M_X}^{t-1}\Big),$$

where $\hat{f}$ denotes a trained mapping from historical functional–vector sequences to the space of functions $\mathcal{F} = \{g : \mathcal{T} \mapsto \mathbb{R}\}$. The look-back horizons $M_Y$ and $M_X$ may differ depending on how functional and auxiliary data are incorporated, with $M = \max(M_Y, M_X)$. The model is trained using samples $(\{(Y_s, X_s)\}_{s=t-M}^{t-1}, Y_t)$ collected at $t \in \{t_1, \ldots, t_N\}$.

This setting generalizes classical functional time series prediction, which relies solely on $\{Y_t\}$, by introducing synchronized covariates $\{X_t\}$. Three structural features characterize the problem:

**Functional structure:** the smoothness and dependence of $Y_t(\cdot)$ across the continuous domain $\mathcal{T}$,

**Temporal dynamics:** the dependence of each $Y_t$ on historical functional trajectories $\{Y_s\}_{s=t-M_Y}^{t-1}$,

**Cross-modal interactions:** the influence of auxiliaries $\{X_s\}_{s=t-M_X}^{t-1}$ on the evolution of $Y_t$.

The central challenge is to design models that faithfully integrate these aspects to achieve accurate and robust forecasting of future functional observations.

**Motivating Example: Yield Curve Forecasting.** In bond markets, a central object of interest is the yield curve, or term structure of interest rates, which relates bond yields to their maturities. Its shape reflects expectations of economic growth, inflation, and monetary policy, making it indispensable for pricing fixed-income securities, managing portfolios, and guiding policy decisions. Central banks and market data providers therefore publish benchmark yield curves daily, forming a functional time series indexed by time-to-maturity. Yet, yield curves are not determined in isolation: their construction and dynamics are shaped by bond-level information. Two categories are particularly relevant. *Market transaction variables*, such as trading volume, bid–ask spreads, and bond prices, capture liquidity conditions and risk perceptions at different maturities. *Bond-specific characteristics*, including coupon rate, issuance date, time to maturity, and credit rating, directly influence how individual bonds contribute to curve construction. These features naturally form a time

series of vector sets. Since the number of bonds being traded varies across time, the size of such set is not constant. In practice, policy banks and treasuries explicitly rely on such bond-level data when constructing benchmark curves. It is therefore natural to leverage this information for forecasting.

**Broader Context.** This formulation extends beyond finance. In energy systems, daily electricity load curves (functions of time of day) are shaped by weather variables across locations. In climate science, temperature or pollution concentration profiles are recorded as daily functions, influenced by exogenous drivers such as emissions or atmospheric conditions. Across domains, the task of forecasting functional time series with synchronized vector inputs captures an important and recurring data structure, making it a compelling target for new representation learning methods.

## 3 METHODOLOGIES

To address the three challenges outlined above, i.e., (i) capturing functional structure, (ii) modeling temporal dynamics, and (iii) fusing functional and vector-valued information, we investigate a collection of candidate techniques along each dimension. Every model can be viewed as a combination of choices across these axes, yielding a family of architectures that allows us to systematically assess the contribution of each component. Below, we present the details of each approach.

### 3.1 FUNCTIONAL REPRESENTATION ALONG THE SPATIAL DOMAIN

The first design choice concerns how to represent the functional observation $Y_t$. Since $Y_t(\tau)$ is defined over a continuous domain $\mathcal{T}$, the representation must capture both smoothness and cross-sectional dependencies. We consider four functional representations:

**Discretization** (D). The function is sampled on a fixed grid $\{\tau_1, \ldots, \tau_C\}$, producing the $C$-dimensional vector $(Y_t(\tau_1), \ldots, Y_t(\tau_C))$. This simple baseline is easy to implement but may not generalize across different resolutions.

**Parameterization** (P). The function is represented using a small number of coefficients, either from a parametric family (e.g., Nelson–Siegel curves in yield modeling) or from a basis expansion (e.g., splines or Fourier). This reduces dimensionality and enforces smoothness.

**Neural-functional-encoder** (N). Building on the framework of functional neural networks (FNNs) Thind et al. (2023), we replace the conventional dot product in the first hidden layer with an integral operator that respects the functional structure. Let $\beta(\tau)$ be a learnable functional weight expanded as $\beta(\tau) = \sum_{i=1}^{m} c_i \phi_i(\tau)$. Then the first-layer operation becomes $\int_{\mathcal{T}} \beta(\tau) y(\tau) \, d\tau = \sum_{i=1}^{m} c_i \left( \int_{\mathcal{T}} \phi_i(\tau) y(\tau) \, d\tau \right)$, which reduces to a standard dot product between coefficients $(c_i)$ and basis-projected covariates. This provides a flexible, nonparametric mechanism to encode functions while preserving their intrinsic structure.

**Functional Transformer** (F). We extend the Transformer architecture to operate directly on functional inputs. Rather than discretizing $Y_t$ into a fixed vector, attention computations (queries, keys, and feedforward layers) are redesigned to respect the continuous nature of the input. This enables richer interactions across the spatial domain and allows the model to exploit smoothness and continuity. Owing to the complexity of this design, detailed formulations are deferred to Section 4.

### 3.2 DYNAMIC REPRESENTATION ALONG THE TEMPORAL INDEX

The second design choice is how to capture the dependence of $Y_t$ on its history $\{Y_s\}_{s=t-M_Y}^{t-1}$. We explore two approaches:

**Non-sequential model.** Lagged functional summaries (e.g., discretized values or basis coefficients of $Y_{t-1}, Y_{t-2}, \ldots$) are concatenated with covariates $X_t$ and treated as static predictors, effectively reducing the temporal dimension to a regression-style problem. Under this formulation, a variety of supervised learning models can be applied; in our study, we adopt a multi-layer perceptron (MLP) as the representative approach for this treatment.

**Sequential model.** Recurrent neural networks (RNNs) or attention-based architectures, i.e., Transformers (TFs), update hidden states sequentially to capture dynamics across time. In our experi-

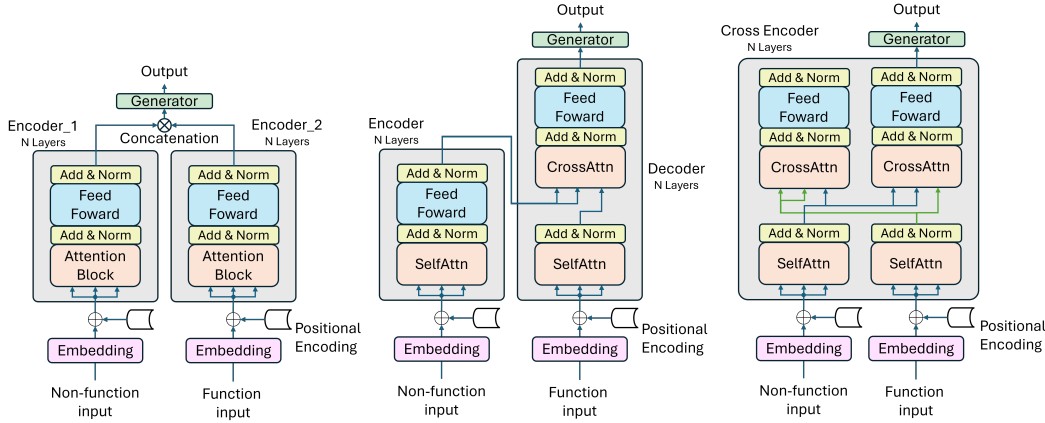

Figure 1: Illustration of Bi-encoder (left), Encoder-decoder (middle), and Cross-encoder (right).

ments, TFs consistently outperformed RNNs such as LSTMs in both predictive accuracy and computational efficiency, and are therefore adopted as our primary sequential model.

### 3.3 Representation Fusion of Functional and Irregular Information

The third design choice is how to integrate functional observations $\{Y_t\}$ with vector-valued covariates $\{X_t\}$. We consider four fusion strategies:

**Direct mapping** (DM). Vector covariates are projected into the same latent space as functional representations and then combined by concatenation or addition. When covariates correspond to function samples at specific domain points, they can alternatively be mapped into functional form through interpolation, basis expansion, or parametric curve fitting (e.g., Nelson–Siegel in our experiments for yield curve modelling). This aligns modalities at the functional level but risks discarding components of $X_t$ that cannot be faithfully mapped.

**Bi-encoder** (BE). Functional and vector modalities are encoded separately, and their latent representations are combined through pooling or interaction layers. This preserves modality-specific structure but limits direct cross-modal communication during feature extraction, potentially overlooking fine-grained dependencies between the functional and auxiliary vector inputs.

**Encoder–decoder** (ED). One modality (e.g., vectors) is encoded and used to condition the decoding of the other (e.g., functions). This asymmetric design naturally captures directional dependencies.

**Cross-encoder** (CE). Functional and vector data are concatenated at the input level and processed jointly through a single encoder, enabling early cross-modal interaction. Cross-attention extensions allow fine-grained alignment across modalities. While powerful, this tight coupling may also propagate noise from one modality to the other if irrelevant features dominate.

A schematic illustration of the architecture of BE, ED and CE is provided in Figure 1. To adopt them, one has to use TF as the base architecture.

For clarity, the proposed neural model's names are constructed by combining the choices across the three methodological dimensions (temporal dynamics, intra-functional representation, and fusion strategy); the full list of model variants is summarized in Appendix A.

## 4 Functional Transformer

This section presents an attention-based architecture for functional time series prediction with irregular vector covariates. As foreshadowed in Section 3.1, the design unifies the advanced strategies for all three challenges within a single framework. The model jointly learns temporal dependencies in function-valued data and contextual signals from variable-sized sets of vectors.

### 4.1 GENERALIZED ATTENTION WITH FUNCTIONAL INPUTS

To jointly process functional and vector inputs, we generalize Transformer attention to accept function-valued tokens. The central idea is to treat functions as infinite-dimensional objects and project them into finite-dimensional embeddings for query, key, and value computations. This allows attention to be computed in the standard way while maintaining compatibility with functional data structures. We begin with reviewing the standard (scaled) dot-product attention. For vector tokens $x_i \in \mathbb{R}^{d_{\text{in}}}$, the query, key, and value vectors are obtained via learned linear projections: $Q_i = W_Q x_i$, $K_j = W_K x_j$, and $V_j = W_V x_j$ with attention weights and outputs computed as

$$\alpha_{ij} = \frac{\exp(Q_i \cdot K_j)}{\sum_k \exp(Q_i \cdot K_k)}, \qquad \text{Attn}(x_i) = \sum_j \alpha_{ij} V_j. \tag{1}$$

For functional inputs $Y \in \mathcal{F} = \{g : \mathcal{T} \mapsto \mathbb{R}\}$, we define learnable embeddings $\mathcal{Q}, \mathcal{K}, \mathcal{V} : \mathcal{F} \mapsto \mathbb{R}^d$, which project functions into finite-dimensional queries, keys, and values. This enables three attention types used in this architecture:

**Functional self-attention.** For a sequence $\{Y_j\}_{j=1}^L$, set $Q_i = \mathcal{Q}(Y_i)$, $K_j = \mathcal{K}(Y_j)$, $V_j = \mathcal{V}(Y_j)$ for all $i, j \in \{1, \ldots, L\}$. The $\text{Attn}(Y_i)$ captures time-dependencies across functions.

**Vector-to-functional cross-attention.** For a vector token $x$ attending to functions $\{Y_j\}_{j=1}^L$, set $Q_x = W_Q x$ and use $\mathcal{K}(Y_j), \mathcal{V}(Y_j)$ as keys/values to extract contextual information from functional trajectories.

**Functional-to-vector cross-attention.** For a function $Y$ attending to vector tokens $\{x_j\}$, set $Q_Y = \mathcal{Q}(Y)$ and $K_j = W_K x_j$, $V_j = W_V x_j$ to incorporate fine-grained vector context into functional representations.

Once all inputs are embedded in $\mathbb{R}^d$, attention is computed via equation 1. This unified attention formulation enables flexible and expressive interaction across heterogeneous data types, while preserving the mathematical and computational structure of Transformer architectures. The functional mappings $\mathcal{Q}, \mathcal{K}, \mathcal{V}$ are implemented using specialized functions described in subsection 4.2.2.

### 4.2 IMPLEMENTATION DETAILS

Our framework handles both functional and vector time series. For practical computation, functional objects are approximated with finite representations while vector inputs are embedded into a shared latent space.

**Functional representation.** Each $Y_t$ is represented by evaluations on a fixed grid $\{\tau_1, \ldots, \tau_C\} \subset \mathcal{T}$, yielding a dense vector $Y_t \mapsto [Y_t(\tau_1), \ldots, Y_t(\tau_C)] \in \mathbb{R}^C$. We adopt the same grid as the raw data in experiments. Although our conceptual design treats functions naïvely, this discretized view enables efficient attention while preserving functional structure.

**Vector-set embedding.** At time $t$, the set $X_t = \{x_t^{(i)}\}_{i=1}^{n_t}$ is embedded elementwise via a shared map $\phi$ to produce $X_t' = \{\phi(x_t^{(i)})\}_{i=1}^{n_t} \subset \mathbb{R}^d$. The non-functional input to the temporal encoder is the concatenation of the past $M$ embedded sets in order: $\mathbf{X}_{(t-M):(t-1)} = [X_{t-M}' \| X_{t-M+1}' \| \cdots \| X_{t-1}']$, where $\|$ denotes vector concatenation. This construction preserves the temporal structure, which is then exploited by attention or masking mechanisms in the temporal encoder to handle variable set sizes. In this work $\phi$ is a linear layer to $\mathbb{R}^d$.

#### 4.2.1 FUNCTIONAL POSITIONAL ENCODING

In Transformers Vaswani et al. (2017), positional encodings are added to token embeddings to inject order information. The canonical sinusoidal encoding is defined along two dimensions: the sequence index and the latent feature dimension. This construction is well-suited to sequences of discrete tokens, but does not directly extend to our setting where each input element is itself a function.

A naïve adaptation is to *flatten each function into a long vector* and apply conventional 1D positional encodings along this extended sequence. While straightforward, this approach discards the natural

separation between temporal order and functional order, leading to unnecessarily long sequences and inefficient attention. Another possibility is to *learn intra-functional embeddings directly* and concatenate them with temporal encodings. This allows more flexibility but requires significantly more parameters and risks overfitting, particularly when data are limited.

To better exploit the problem structure, we reinterpret positional encoding as a *two-dimensional mapping* over both sequence index $t$ and functional domain position $\tau \in \mathcal{T}$. Specifically, we define a joint sinusoidal encoding which is added to the input..

$$\mathrm{PE}(t, \tau) = \bigoplus_{k=1}^{C/2} \left[ \sin\left(\omega_k^t\, t + \omega_k^\tau\, \tau\right),\ \cos\left(\omega_k^t\, t + \omega_k^\tau\, \tau\right) \right],$$

where $\{\omega_k^t\}$ and $\{\omega_k^\tau\}$ are frequency scales governing variation across the sequence and intra-functional directions, respectively, and $\oplus$ denotes concatenation. This construction injects structured positional variation simultaneously across the temporal and functional dimensions, preserving the two-level hierarchy of our data while remaining compatible with standard attention mechanisms.

In many applications, the accompanying vector covariates also contain descriptors of what the functional values represent (e.g., instrument identifiers) or of the functional domain itself (e.g., maturity or spatial location). We apply the same 2D encoding scheme to these covariates, embedding them in a space aligned with the functional inputs. This ensures that the Transformer can exploit correspondences across modalities and capture richer cross-modal dependencies.

### 4.2.2 ATTENTION BLOCK IMPLEMENTATION

Attention blocks contain self- and cross-attention modules (Section 4.1), layer normalizations, and feedforward layers, with residual connections. The key innovation lies in generalizing the extraction of query, key, and value representations from functional inputs to vector embeddings.

**Vector tokens.** For embedded vectors $\phi(x_t^{(i)}) \in \mathbb{R}^d$, we use conventional linear projections:

$$Q = W_Q' \phi(x_t^{(i)}), \quad K = W_K' \phi(x_t^{(i)}), \quad V = W_V' \phi(x_t^{(i)}),$$

with $W_Q', W_K', W_V' \in \mathbb{R}^{d \times d}$, as in classical attention mechanisms.

**Functional tokens.** For discretized functional inputs $Y \in \mathbb{R}^C$, the query, key, and value representations are obtained via independent nonlinear mappings:

$$Q = \mathcal{Q}(Y), \quad K = \mathcal{K}(Y), \quad V = \mathcal{V}(Y), \qquad \mathcal{Q}, \mathcal{K}, \mathcal{V} : \mathbb{R}^C \mapsto \mathbb{R}^d.$$

To transform functional sequences into compact vector representations, we adopt a convolutional encoding scheme. The encoder treats each function as a one-dimensional signal and applies successive convolutional and pooling operations to extract hierarchical features.

Specifically, the input sequence (viewed as a single–channel signal) is first processed by a one–dimensional convolution layer with $d$ output channels. This step projects the raw functional input into a higher–dimensional feature space, enabling the model to capture multiple local patterns simultaneously. Increasing the number of channels allows the encoder to disentangle diverse structural components of the curve (e.g., local fluctuations, slope variations) that a single channel cannot adequately represent. Then a second convolution layer is applied, keeping the number of channels fixed and allowing the model to refine and re–combine the extracted local features without introducing unnecessary parameter growth. This design stabilizes the feature space while deepening the receptive field, ensuring that both short– and medium–range dependencies within the functional sequence are effectively encoded. Following the convolution blocks, a global average pooling operation is applied along the temporal dimension. Pooling serves two purposes: (i) it aggregates variable–length functional inputs into a fixed–size representation, and (ii) it enforces invariance to small temporal shifts or local noise. As a result, the output is a compact hidden representation that summarizes the global characteristics of the curve. Finally, a linear mapping is used to project the pooled representation into the latent embedding space of dimension $d$.

**Shape alignment.** Finally, the output of the attention block is passed through a linear layer to match the shape of the attended data: for vector inputs, the output dimension is $d$, while for functional inputs, the output is reshaped to match the discretization length $C$. This step ensures that the attention output aligns with the shape of the input data, which is necessary because the attention block is followed by a residual connection, as in the classic Transformer architecture.

### 4.2.3 OUTPUT GENERATOR

The final predictor maps the last functional state to a function $\hat{Y}_t$. Let $z \in \mathbb{R}^d$ denote the final functional embedding for time $t-1$ from the encoder/decoder stack. We define $\hat{Y}_t(\tau) = g(z, \tau)$, where $g$ is implemented as a shallow network in $(z, \tau)$ followed by a Gaussian kernel smoother. This produces a smooth, coherent function on $\mathcal{T}$ aligned with the structure of the original observations.

## 5 EXPERIMENTAL SETUP AND RESULTS

We evaluate our framework using a real-world financial dataset of China Development Bank (CDB) bonds. The raw data are sourced from two official platforms: the China Foreign Exchange Trade System (CFETS) and ChinaBond. Our dataset spans from January 2021 to August 2025 at daily frequency, yielding 1,160 trading days in total. We use the first $N = 600$ days (Jan 2021–May 2023) for initial training and the remainder for testing and re-training; see the training and testing scheme detailed in subsection 5.1. For each trading day, two types of data are collected:

**Functional data** $(Y_t(\tau), \tau \in [0, 30])$**.** The benchmark yield curve published by China Central Depository & Clearing Co., Ltd. (CCDC), representing the term structure of interest rates. Each curve is linearly interpolated and discretized on $[0, 30]$ (years) at 0.05-year increments, with additional evaluation points at $(0.08, 0.13, 0.18, 0.23)$ to capture short-end behavior, yielding $C = 205$ points per curve.

**Vector data** $(X_t = \{x_t^{(1)}, \ldots, x_t^{(n_t)}\})$**.** The cross-section of all traded CDB bonds on the same day, including closing prices, trade volume, quote volume, and bond-level metadata such as time to maturity and coupon rate, which form a vector $x_t^{(i)} \in \mathbb{R}^m$. As the set of traded bonds varies by day, this forms a time series of sets with heterogeneous cardinality $(n_t)$.

Although our data were collected on daily basis, we formulate our forecasting task with weekly curves for their stationarity and practicality because high-frequency (e.g., daily) data are very noisy while low-frequency (e.g., monthly) forecasts are hard to generate actionable insights for bond pricing and trading decisions. Hence, our time index $t$ uses a week as one unit but the data $(\{\{Y_s\}_{s=t-5}^{t-1}, X_{t-1}\}, Y_t)$ are sampled from the trading days, forming still $N$ rows for the training.

**Task definition.** Given the historical sequence of weekly yield curves from $t-5$ to $t-1$ ($M_Y = 5$) and the bond-level vector set on the last day of week $t$ at or before 16:55 (deemed as market data at $t-1$, i.e., $M_X = 1$), the goal is to predict the yield curve for week $t$ ($\hat{Y}_t$) at 16:55. Predictions are evaluated against the benchmark yield curve officially published by CCDC at 17:00 at $t$ ($Y_t$).

### 5.1 TRAINING AND TESTING SCHEME

We adopt a rolling-window procedure to mimic live deployment and to ensure training always uses recent data while avoiding look-ahead:

**Training phase.** During the training, the model is updated using overlapping windows of length $N = 600$. For each window, a set of indices is generated to extract 16-sized sequential batches of functional and vector covariates. In order words, the new data come in the last batch, which keeps the model up-to-date. The model is trained for 500 epochs per batch.

**Testing phase.** In the testing segment, we employ a moving-window forecast strategy. At each step, the model is either fine-tuned on the window ending at $t-1$ or reused directly, and then a prediction is generated for the yield curve at $t$. The process then advances by one week, and the window is rolled forward until the end of the test sample. Predictions from all windows are concatenated to form the full out-of-sample forecast sequence.

This setup avoids look-ahead bias and ensures each prediction conditions on the latest realized information. Unlike recursive autoregression, our forecasts at $t$ condition on the observed $Y_{t-1}$, preventing error accumulation across weeks.

## 5.2 BASELINES AND PERFORMANCE METRIC

In addition to the model variants summarized in Appendix A (Table 2), we benchmark against classical approaches used in yield curve modeling:

**Parametric Nelson–Siegel** (`NS`). A cross-sectional `NS` model fit to observed bond prices at $t$.

**Dynamic Nelson–Siegel** (`Dym-NS`). A dynamic extension of `NS`: parameters are estimated from recent yield curves $(t - M_Y, \ldots, t - 1)$ and a vector autoregressive (VAR) model is fit to the parameter series to forecast day-$t$ parameters; see Caldeira et al. (2025) for a review.

**Nonparametric B-Spline** A `B-spline` fit of the yield curve from observed bond prices at $t$.

**Dynamic B-Spline** (`Dym-B-Spline`) Analogous to the extension of `NS`, dynamic `B-Spline` models the coefficients of past B-splines with VAR to forecast the ones for time $t$.

We introduce two performance metrics to evaluate our proposed models and benchmarks:

**Root mean squared error (RMSE)** calculates over maturities, averaged across test sequences:

$$\text{RMSE} = \sqrt{\frac{1}{C|\mathcal{D}_{\text{test}}|} \sum_{t \in \mathcal{D}_{\text{test}}} \sum_{c=1}^{C} \left(\hat{Y}_t(\tau_c) - Y_t(\tau_c)\right)^2}.$$

RMSE emphasizes both level and shape fidelity across the maturity grid.

**Average max error (AME)** calculates over maturities, averaged across test sequences:

$$\text{AME} = \frac{1}{C} \sum_{c=1}^{C} \max_{t \in \mathcal{D}_{\text{test}}} \left|\hat{Y}_t(\tau_c) - Y_t(\tau_c)\right|.$$

AME emphasizes the worst-case deviation at each maturity, thereby capturing the model's robustness in preventing large errors at specific points on the yield curve.

## 5.3 EXPERIMENTAL RESULTS

We evaluate the proposed framework and all model variants in Appendix A under both *absolute* and *delta* settings. Because different classes treat functional inputs differently (e.g., discretization, parameterization, functional encoders, functional attention), functional inputs are preprocessed into the representation appropriate to each class, following Section 3.1. Unless noted otherwise, the same training/testing scheme is used across models. Table 1 summarizes our experimental results.

**Interpretation.** Three patterns emerge from Table 1. *(i) Functional representation choice.* Within the non-sequential family, the neural functional encoder (`MLP-N-DM`, RMSE 0.0315) and discretization (`MLP-D-DM`, 0.0316) are the strongest, while parameterization is clearly weaker (`MLP-P-DM`, 0.0822). In the sequential family, discretization and neural functional encoders again outperform parameterization (e.g., `TF-D-DM` 0.0374, `TF-N-DM` 0.0399 vs. `TF-P-DM` 0.0431). *(ii) Fusion strategy and robustness.* Direct mapping (DM) remains the most reliable way to incorporate irregular vector sets: for comparable functional choices, DM yields the best or near-best Transformer results (e.g., `TF-D-DM` vs. `TF-D-BE`/`CE`/`ED`). Notably, Transformer variants exhibit *stable* performance across representations and fusion strategies (RMSE $\approx$ 0.037-0.047, AME $\approx$ 0.300-0.318), underscoring robustness even when not the absolute top performer. *(iii) Classical baselines.* Modern learning approaches dominate traditional fits on this weekly yield curve forecasting task: dynamic B-splines and Nelson–Siegel variants lag considerably (e.g., `Dym-B-Spline` 0.0863; `NS` 0.0778; `Dym-NS` 0.1300), and direct B-spline fits remain unreliable (RMSE $> 1$). Beyond averages, the error profile is informative: `MLP-N-DM` achieves the best AME (0.245), while Transformers cluster tightly around $\sim 0.30$, indicating fewer large misspecifications across weeks. This "tight band" behavior is visible across all fusion variants for Transformers, suggesting that their inductive bias yields consistent performance irrespective of the specific integration strategy for irregular sets.

Table 1: Yield curve forecasting performance. *Remarks: For the `Dym-NS` model, performance metrics are reported after excluding the 10 problematic forecasts (outliers). For the `B-spline` model, the metrics remain above 1 even after excluding 100 "outliers". Other models did not have any "outliers" or non-converged cases.

| Model | RMSE | AME |
|---|---|---|
| B-Spline | >1* | >1* |
| Dym-B-Spline | 0.0863 | 0.299 |
| NS | 0.0778 | 0.336 |
| Dym-NS | 0.1300* | 0.283* |
| MLP-D-DM | 0.0316 | 0.251 |
| MLP-P-DM | 0.0822 | 0.276 |
| MLP-N-DM | 0.0315 | 0.245 |
| TF-D-DM | 0.0374 | 0.301 |
| TF-P-DM | 0.0431 | 0.306 |
| TF-N-DM | 0.0399 | 0.306 |
| TF-D-BE | 0.0414 | 0.307 |
| TF-P-BE | 0.0424 | 0.311 |
| TF-N-BE | 0.0407 | 0.318 |
| TF-F-BE | 0.0472 | 0.300 |
| TF-D-CE | 0.0412 | 0.305 |
| TF-P-CE | 0.0427 | 0.301 |
| TF-N-CE | 0.0427 | 0.315 |
| TF-F-CE | 0.0472 | 0.305 |
| TF-D-ED | 0.0406 | 0.314 |
| TF-P-ED | 0.0406 | 0.309 |
| TF-N-ED | 0.0424 | 0.315 |
| TF-F-ED | 0.0469 | 0.308 |

Two additional observations provide context. First, while the best MLPs achieve the lowest RMSEs, they also display a *wider spread* across configurations (from 0.0315 to 0.0822), whereas Transformers are comparatively uniform across choices, a desirable property when stability across design decisions is important. In practice, this translates to less retuning burden and more predictable behavior when swapping representations or fusion blocks, which is crucial for production deployment. Second, functional encoders that learn from curves end-to-end (`MLP-N`, `TF-N`) consistently outperform fixed parameterizations, supporting the representation-learning premise of our framework. We also note that the "functional Transformer" variants (`TF-F`) are competitive and stable but slightly behind `TF-D/N` counterparts in this configuration, pointing to a tuning gap rather than a conceptual limitation. Overall, these results favor learning functional structure directly, using simple fusion for irregular sets, and leveraging Transformer backbones for robustness across modeling variants.

## 6 CONCLUSION

We presented a unified framework for forecasting functional time series with irregular vector covariates and instantiated it with a functional Transformer. Our experiments on *weekly* yield-curve forecasting show that end-to-end functional learning (neural functional encoders or functional attention) is competitive with, and often superior to, fixed parameterizations; that lightweight fusion (direct mapping) is more reliable than early or asymmetric cross-modal coupling; and that Transformer backbones deliver *stable* performance across representations and fusion strategies, even when the very best RMSE is achieved by MLPs.

Meaningful adjustments can concentrate in local maturity regions even when curve-level shifts are modest. To reflect economic salience, we advocate a *within-curve-weighted RMSE* that emphasizes actively traded maturities (e.g., weights proportional to trading volume). Our functional Transformer remains under active tuning; updated results will be posted to the project repository. Given its inductive bias and flexibility, we expect further gains with larger datasets and careful fine-tuning, and see promising extensions to reinforcement learning and function-conditioned generative modeling.

## 7 REPRODUCIBILITY STATEMENT

To facilitate reproducibility, we provide a complete implementation of our framework at `https://anonymous.4open.science/r/ICLR2026_Sub23115-25EF/`. The package includes all model codes together with the hyperparameter settings used for each baseline and proposed model. A consolidated dataset, along with detailed preprocessing scripts and instructions, is also included in the repository at the link above.

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

## A  MODEL VARIANTS

Table 2 summarizes the full set of model variants constructed by combining design choices across the three methodological dimensions. Each model name is formed by concatenating the base architecture (MLP or TF) with its respective functional representation and fusion strategy. For example, `TF-F-BE` denotes a Transformer-based sequential model with a functional-transformer representation and a bi-encoder fusion design.

Table 2: Summary of model variants across methodological dimensions.

| Model | Temporal dynamics | Intra-functional structure | Fusion strategy |
|---|---|---|---|
| MLP-D-DM | Non-sequential model | Discretization | Direct Mapping |
| MLP-P-DM | Non-sequential model | Parameterization | Direct Mapping |
| MLP-N-DM | Non-sequential model | Neural-functional-encoder | Direct Mapping |
| TF-D-DM | Sequential model | Discretization | Direct Mapping |
| TF-P-DM | Sequential model | Parameterization | Direct Mapping |
| TF-N-DM | Sequential model | Neural-functional-encoder | Direct Mapping |
| TF-D-BE | Sequential model | Discretization | Bi-Encoder |
| TF-P-BE | Sequential model | Parameterization | Bi-Encoder |
| TF-N-BE | Sequential model | Neural-functional-encoder | Bi-Encoder |
| TF-F-BE | Sequential model | Functional-transformer | Bi-Encoder |
| TF-D-CE | Sequential model | Discretization | Cross-Encoder |
| TF-P-CE | Sequential model | Parameterization | Cross-Encoder |
| TF-N-CE | Sequential model | Neural-functional-encoder | Cross-Encoder |
| TF-F-CE | Sequential model | Functional-transformer | Cross-Encoder |
| TF-D-ED | Sequential model | Discretization | Encoder-Decoder |
| TF-P-ED | Sequential model | Parameterization | Encoder-Decoder |
| TF-N-ED | Sequential model | Neural-functional-encoder | Encoder-Decoder |
| TF-F-ED | Sequential model | Functional-transformer | Encoder-Decoder |

