# OpenReview forum: "A Unified Representation Learning Framework for Functional, Temporal, and Irregular Data"
_ICLR.cc/2026/Conference — Submitted to ICLR 2026_

### Official Review · Reviewer_fn8U · 2025-10-30

**Soundness:** 2
**Presentation:** 2
**Contribution:** 3
**Rating:** 4
**Confidence:** 4

**Summary:**

This paper introduces a novel Transformer-based framework for functional data representation learning. The core innovation lies in its capacity to effectively process function-valued inputs characterized by temporal dynamics and irregular inputs.

The proposed architecture first employs a time-sensitive self-attention module to project the temporal, continuous functional inputs into concise, finite-dimensional vector representations. Following this, the paper investigates four distinct representation fusion strategies specifically designed to aggregate or harmonize information derived from the irregularly sampled components of the data.

The experimental section rigorously explores various combinations of the introduced module designs and fusion techniques. Overall, the methodology presents a promising direction for handling complex functional data streams.

**Strengths:**

1. The problem formulation is exceptionally clear and well-justified. The authors effectively summarize the key challenges inherent in handling temporal functional inputs and have compiled a comprehensive set of techniques aimed at addressing these issues. The paper's strength lies in its systematic and rigorous approach to the design space, covering:

   - Functional Representations: Analysis of how the continuous inputs are effectively digitized or summarized.

   - Representation Learning Architectures: Exploration of model structures suitable for capturing temporal dependencies.

   - Modality Fusion Strategies: Investigation of methods to integrate different data types or information sources.

2. A significant contribution is the validation of the proposed methods on a real-world production dataset from the China Development Bank (CDB). This inclusion strongly demonstrates the framework's practical efficacy and its potential for real-world deployment in solving complex production-level problems.

**Weaknesses:**

1. The proposed method in this paper has not been tested on an open-source benchmark, which makes it unclear whether the experimental results are sufficient.
2. Furthermore, the presentation of only one experimental result table for the entire methodology is inadequate for rigorously selecting the best framework configuration. Therefore, the authors should consider adding experiments in other domains or research topics to demonstrate the method's generalizability.

**Questions:**

1. Will the benchmark dataset utilized for the experiments be made open-source?

---

### Official Review · Reviewer_193b · 2025-10-31

**Soundness:** 2
**Presentation:** 2
**Contribution:** 2
**Rating:** 2
**Confidence:** 3

**Summary:**

The authors pose the paper as a unified representation learning framework which can handle functional data, temporal structure, and irregular vector sets. They test many different ways of encoding functional inputs to fit in with a transformer network, and test it on the yield curve.

**Strengths:**

First time a yield curve has been adopted in the form of observations rather than discrete data points. Going over some pretty realistic scenarios with the financial data.

They give concrete implementational details and provide the code for the experiments.

They seem to have strong results against the baselines that they implemented.

**Weaknesses:**

No ablation done on the positional encoding actually being better than a simpler method such as linear projection, but it is claimed as being principled.

The others mention papers Dupont et al.(2022); Zhou et al. (2023); Tran et al. (2025), but evaluations are only done on some test spline-based models and functional MLP, leaving there a lot to be desired with the baseline comparisons.

The authors claim that the functional transformer is section 4.1 to operate directly on functional inputs. However, in 4.2.2, this seems to be implemented by discretizing the curve on a fixed grid, running a 1D conv layer, giving it a 2D positional encoding t and tau, and then adopting vanilla attention blocks. Novelty-wise, it still seems to be a form of a sequence-based transformer, not an analytically continuous operator.

Results are evaluated on one dataset but claimed to be broadly applicable; some more evidence with more experiments should be included to support this claim.

The rolling style of training and evaluating seems like it would be quite expensive to deploy in the real world

**Questions:**

If the authors can clarify how exactly their model is doing functional propagation across the attention layers (see weaknesses for the confusion), that would be great.

---

### Official Review · Reviewer_8SyS · 2025-11-01

**Soundness:** 2
**Presentation:** 1
**Contribution:** 2
**Rating:** 2
**Confidence:** 5

**Summary:**

This paper proposes a unified framework to learn representations from data that is simultaneously functional (continuous), temporal (sequential), and irregular (unstructured vector sets). The authors motivate this problem using yield curve forecasting, where one must predict a future curve (a function) based on a history of past curves (functional time-series) and a set of bond-level trades (irregular vector set).

The paper's proposed solution culminates in a "Functional Transformer" (F) architecture (Section 4), which claims to generalize attention to operate directly on continuous, function-valued inputs. The authors test this and other variants (using MLPs, standard Transformers, and different encoders/fusion strategies) on a yield curve forecasting task, claiming "consistent gains over classical econometric models."

Despite the interesting problem formulation, the paper is fundamentally flawed. It suffers from two problems:
1.  The central methodological claim of a "continuous functional Transformer" is directly contradicted by the implementation, which is a standard discretization.
2.  The experimental results (Table 1) show that the paper's main proposed architecture (the Transformer) is significantly *outperformed* by a simple, non-sequential MLP baseline, invalidating the paper's core premise.

Furthermore, the experiments are poorly described, the appendix is incomplete, and no ablation studies are provided to justify the "unified" framework.

**Strengths:**

The paper has two redeeming qualities:

1.  **Interesting Problem Formulation:** The paper identifies and formalizes a challenging and important problem: how to learn from data that is simultaneously functional, temporal, and irregularly structured. This problem is highly relevant in many domains beyond the paper's motivating example.
2.  **Systematic Taxonomy:** The breakdown of the problem into three design axes in Section 3 (Functional Representation, Temporal Dynamics, and Representation Fusion) is a clear and useful conceptual framework for organizing and thinking about potential solutions.

**Weaknesses:**

### 1. The "Functional Transformer" is Not Functional
The paper's entire novelty rests on the claim of a new architecture that "extend[s] attention mechanisms to handle entire functions along a continuous axis" (Abstract) and "generalize[s] self- and cross-attention to operate on function-valued inputs" (Contribution 2). Section 4.1 details this "Generalized Attention" by defining query, key, and value as projections from the function space $\mathcal{F}$ to $\mathbb{R}^d$. This claim is false.

The implementation, as described in Section 4.2 ("Implementation Details"), is a simple, standard discretization:

> "Each $Y_t$ is represented by evaluations on a fixed grid... yielding a dense vector... We adopt the same grid as the raw data... Although our conceptual design treats functions naıvely, this discretized view enables efficient attention..."

This is a bait-and-switch. The paper promises a novel continuous-domain attention mechanism but delivers the most common baseline (Discretization, 'D'). The "Functional Transformer (F)" variants tested in Table 1 (e.g., `TF-F-BE`) are simply a 1D-CNN encoder applied to the discretized vector, followed by a standard Transformer. This is a common bi-encoder design, not a novel attention mechanism. The core methodological contribution of the paper does not exist.

### 2. The Proposed Model (Transformer) Fails

The paper spends its entire methodology section (Section 4) building up the "Functional Transformer" as the key contribution. The premise is that a "Sequential model" (like a Transformer) is necessary to "capture dynamics across time" (Section 3.2). The paper's own results (Table 1) refute this.
* The best-performing model is `MLP-N-DM` (RMSE **0.0315**).
* The second-best model is `MLP-D-DM` (RMSE **0.0316**).

Both of these are MLPs, which the paper defines as "Non-sequential models." The best-performing *Transformer* model (`TF-D-DM`) achieves an RMSE of **0.0374**, which is **~19% worse** than the simple MLP baseline.

The "Functional Transformer" (TF-F-*) variants, which supposedly use the CNN encoder, perform even worse (RMSEs of **~0.047**).

The paper's core experimental result is that a simple, non-sequential MLP on discretized data beats the complex, sequential Transformer architecture this paper is built around.  The authors' attempt to spin this in Section 5.3 ("Transformer variants exhibit stable performance") is unconvincing. The MLPs are not only more accurate, but their "spread" (0.0315 to 0.0822) is only wide because the 'P' (Parameterization) method is clearly a bad representation, a fact that *also* holds for the Transformers (0.0374 to 0.0472).

### 3. Incomplete and Confusing Experiments

The experimental design is opaque, and the appendix is critically incomplete.

* **No Hyperparameters:** The Appendix is nearly empty. There are no details on any model hyperparameters. What is the embedding dimension $d$? How many layers and heads in the Transformers? What is the MLP architecture? What is the CNN architecture for the 'F' models? What is the decoder $g(z, \tau)$ architecture? Without this, the work is irreproducible.
* The training scheme is nonsensical. "The model is trained for 500 epochs per batch." This is almost certainly a typo, but it's impossible to know what was actually done. Does it mean 500 epochs per *window*? This lack of clarity is unacceptable.
* **Trivialized Problem (Sec 5):** The paper is motivated by modeling *temporal dynamics* of *irregular sets* (i.e., $\{Y_s\}$ and $\{X_s\}$). However, the experiment only uses **one** irregular set from the previous day ($M_X = 1$). This completely discards the "temporal dynamics" aspect of the irregular data, a key part of the paper's "unified" problem statement.
* **No Ablation Studies:** The paper proposes a "unified framework" for three modalities but provides **zero ablation studies**. The most obvious question is: *does the irregular vector data $X_t$ even help?* What is the performance of `MLP-N` or `TF-D` *without* the "Direct Mapping" (DM) fusion? Without this baseline, it's impossible to know if the cross-modal fusion (a key contribution) provides any value at all.

**Questions:**

1.  Can you please clarify the discrepancy between Section 4.1 (claiming continuous functional attention) and Section 4.2 (stating it's a "discretized view")? Is there any part of the `TF-F` model that is not just a standard 1D-CNN encoder on a fixed grid?
2.  Given that the non-sequential `MLP-N-DM` (RMSE 0.0315) and `MLP-D-DM` (RMSE 0.0316) significantly outperform all sequential Transformer models (best is `TF-D-DM` at 0.0374), how can you justify the paper's focus on the Transformer as a "unified framework"? Do these results not imply that modeling temporal dynamics is unnecessary for this task?
3.  What is the performance of your best models (e.g., `MLP-N-DM` and `TF-D-DM`) when trained only on the functional time-series data $\{Y_s\}$, without the irregular vector set $X_t$? This is a critical baseline to prove that your cross-modal "unified" framework provides any benefit.
4.  The appendix is missing all hyperparameters. Could you please provide the full architecture details (layers, heads, dimensions, CNN kernel sizes, etc.) for all models? Also, could you please clarify the "500 epochs per batch" statement in Section 5.1?
5.  Why was the temporal lookback for the irregular data set to $M_X = 1$? This seems to abandon the "temporal dynamics" aspect of the irregular data, which was a core part of the problem formulation in Section 2.

---

### Official Review · Reviewer_2Ua7 · 2025-11-01

**Soundness:** 2
**Presentation:** 2
**Contribution:** 2
**Rating:** 2
**Confidence:** 3

**Summary:**

This paper presents a unified representation learning framework for functional, temporal, and irregular data, with a focus on yield curve forecasting as a motivating application. The work addresses an important and underexplored problem of integrating heterogeneous data types (functional time series and unordered vector sets) within a single architecture.

**Strengths:**

The paper systematically combines multiple representation learning strategies into a cohesive framework, enabling a structured comparison of design choices.

**Weaknesses:**

1.	The authors claim that the framework is broadly applicable, but only tested on yield curve forcasting
2.	In line 301,307-309, and other places, there has informal indent
3.	Yield curve forcasting is only tested on one dataset, it is not enough to evaluate the algorithm.
4.	There has no ablation studies to evaluate each proposed component

**Questions:**

How does the proposed functional self-attention fundamentally differ from applying a standard Transformer to discretized functional data?

The experiments focus on comparing full model variants but lack granular ablation analysis. For example:
How much do individual components (e.g., cross-attention modules) contribute to performance?
Why do MLPs sometimes outperform Transformers (e.g., MLP-N-DM vs. TF-D-DM)? Is this due to overfitting or inadequate training of Transformer variants?

The paper only evaluates on ​one financial dataset​ (Chinese bond yields). While yield curve forecasting is a valid application, the framework claims generality across domains (energy, climate). Where are the experiments on other functional time series (e.g., electricity load, climate data)?

The comparison to classical models is thorough, but where are the ​modern deep learning baselines​?

---

### Official Review · Reviewer_fr2B · 2025-11-01

**Soundness:** 2
**Presentation:** 2
**Contribution:** 2
**Rating:** 2
**Confidence:** 4

**Summary:**

This paper looks at integrating functional elements into a transformer formalism. At its heart it projects an infinite dimensional functional representation into a finite dimensional space for embedding within a fairly standard transformer like object. It does an experiment testing the ability of the approach.

**Strengths:**

This paper tackles a real issue, and does so in a sensible way, it motivates what it does, and it provides a fairly general approach or set of approachers for handling this. The problem formulation is fine: the paper adds conditioning side information to functional time series.

**Weaknesses:**

There are three key weaknesses: 1) the writing 2) the contribution, 3) the experiments

1) the writing: the paper seems to consist of lots of lists of things that are really not implemented or considered features of the paper. They take the form of "we have these choices", but many of these are not handled apart from being listed. This makes the papers somewhat incoherent. It feels like the authors just thought about what they could write on this topic rather than focusing on the research in question.

2) the contribution: my question is to what this adds beyond what an individual faced with this problem would implement by default? What is the insight? Adding conditioning variables into any problem is pretty standard across most domains, and the approach described seems similar to that which pretty much what anyone would do.

3) the experiments: I ask whether I believe the experiments have demonstrated this is the right way to do this rather than a way to do this, and I feel the experiments do not demonstrate this. The experimental methods being compared against do not really use the same information, there are no ablations on the method, and there is a singular experiment.

Altogether the paper is probably solid for an implementational approach for the problem being tackled, but it is not a research paper. I do not come out with any improved insight or understanding of the field than I had going in.

**Questions:**

Why is this not just what people would do anyway? What is the fundamental insight you bring to this that you claim makes a difference? What should my research take-home be for this paper? What ablations can you demonstrate that your approach has both insight and credence: it is not just one of many methods that would do pretty much the same thing?

---

### Meta-Review · Area_Chair_AiS1 · 2025-12-28

**Summary:**

Reviewers broadly agreed the problem setup is potentially interesting, but they did not find the paper’s claimed technical contribution or empirical support convincing enough for ICLR. Key themes were: a perceived mismatch between the “functional/continuous attention” framing and what is actually implemented, experiments that are too narrow (one dataset, limited modern baselines, no ablations), and insufficient methodological clarity/reproducibility and over-claiming of generality.

**Reviewer Concerns:**

As the authors did not participate in the rebuttal/discussion period, all major concerns remain outstanding.

**Reviewer Scores:**

Given the above, fr2B, 2Ua7, 8SyS, and 193b would likely stay at 2 (reject), while fn8U (initially 4) would likely move down after aligning with the novelty/implementation concerns raised by multiple reviewers.

---

### Decision · Program_Chairs · 2026-01-26

Reject